# Molecular Characteristics and Genetic Evolution of Echovirus 33 in Mainland of China

**DOI:** 10.3390/pathogens11111379

**Published:** 2022-11-18

**Authors:** Wenhui Wang, Huan Fan, Shuaifeng Zhou, Shikang Li, Alitengsaier NIGEDELI, Yong Zhang, Qiang Sun, Yun He, Qin Guo, Xiaoyi Wang, Huanhuan Lu, Jinbo Xiao, Hehe Zhao, Zhenzhi Han, Tianjiao Ji, Le Zhang, Dongmei Yan

**Affiliations:** 1School of Public Health and Management, Shandong First Medical University & Shandong Academy of Medical Sciences, Jinan 250117, China; 2WHO WPRO Regional Polio Reference Laboratory and Ministry of Health Key Laboratory for Medical Virology, National Institute for Viral Disease Control and Prevention, Chinese Center for Disease Control and Prevention, Beijing 102206, China; 3Jiangsu Center for Disease Control and Prevention, Nanjing 210009, China; 4Hunan Center for Disease Control and Prevention, Changsha 410005, China; 5Xinjiang Center for Disease Control and Prevention, Urumqi 830002, China; 6Medical School, Anhui University of Science and Technology, Huainan 232001, China

**Keywords:** echovirus 33, recombinant analysis, genomic characterization, phyletic evolution, spatial transmission

## Abstract

Echovirus, a member of the Enterovirus B (*EV-B*) family, has led to numerous outbreaks and pandemics, causing a broad spectrum of diseases. Based on the national hand, foot, and mouth disease (HFMD) surveillance system, seven strains of echovirus 33 (E33) were isolated from Mainland of China between 2010 and 2018. The whole genomes of these strains were isolated and sequenced, and phylogenetic trees were constructed based on the gene sequences in different regions of the *EV-B* prototype strains. It was found that E33 may be recombined in the *P2* and *P3* regions. Five genotypes (A–E) were defined based on the entire *VP1* region of E33, of which the C gene subtype was the dominant gene subtype at present. Recombinant analysis showed that genotype C strains likely recombined with EV-B80, EV-B85, E13, and CVA9 in the *P2* and *P3* regions, while genotype E had the possibility of recombination with CVB3, E3, E6, and E4. Results of Bayesian analysis indicated that E33 may have appeared around 1955 (95% confidence interval: 1945–1959), with a high evolutionary rate of 1.11 × 10^−2^ substitution/site/year (95% highest posterior density (HPD): 8.17 × 10^−3^ to 1.4 × 10^−2^ substitution/site/year). According to spatial transmission route analysis, two significant transmission routes were identified: from Australia to India and from Oman to Thailand, which the E33 strain in Mainland of China likely introduced from Mexico and India. In conclusion, our study fills the gaps in the evolutionary analysis of E33 and can provide important data for enterovirus surveillance.

## 1. Introduction

Enteroviruses are members of the genus *Enterovirus*, family *Picornaviridae*, and order *Picornaviridae*, which comprise 15 species: *Enterovirus A–L* and *Rhinovirus A–C* [1]. As a member of *Enterovirus B*, E33 is a small, single-stranded, positive-sense RNA virus with a total length of approximately 7495 bp, consisting of a 5′-untranslated region (*5*′*-UTR*), an open reading frame (ORF), a 3′-untranslated region (*3*′*-UTR*), and a poly-A tail [2,3]. The two terminal structures are related to viral translation, replication, and infectivity. The ORF can be cleaved into the structural protein *P1* and the non-structural proteins *P2* and *P3*, which can be further hydrolyzed to the structural proteins *VP1–VP4* and the non-structural proteins *2A–2C* and *3A–3D*, respectively [4]. Specifically, the *VP1* region contains important antigen sites and has a high specificity for different serotypes [5]; thus, a large number of studies have classified genotypes based on this fragment [6,7,8,9]. Enteroviruses most often recombine in the *P2* and *P3* regions, which contain structures that play important roles in RNA replication, protein processing, and the termination of host cell translation [10]. In addition, the virulence and transmissibility of enteroviruses may be altered by genetic recombination [11].

E33 was first discovered in 1959 in Toluca, Mexico. It was then named Toluca-3 during a study involving the mass administration of a live poliomyelitis vaccine. In the 1980s, outbreaks of E33 were reported in Germany, Belgium, and France, with meningitis as the primary symptom [12,13]. Subsequently, it has been sporadically detected in several American countries [14]. In 1994, an outbreak of E33 occurred in a maternity hospital in Japan, where nine neonates were infected and showed severe symptoms, such as encephalitis and disseminated intravascular coagulation [15]. In 2000, an unprecedented outbreak of E33 occurred in New Zealand [16]. The majority of infected patients presented with aseptic meningitis and fulminant hepatitis, resulting in two deaths. A boy who developed flaccid paralysis of the right arm after an E33 infection was also reported [17]. Since the beginning of the 21st century, reports of E33 from Asia and Africa have increased [18,19,20]. E33 was first reported in Yunnan, China, in 2002, isolated from a case of non-poliomyelitis acute flaccid paralysis (NP-AFP) [21]. It is worth noting that in 2013, an influenza-like outbreak caused by E33 occurred in a middle school in Hunan Province, which resulted in 108 students being infected within 21 days [22]. In the same year, E33 was detected in two patients with hand, foot, and mouth disease (HFMD) and meningitis in Yunnan, China.

Although E33 has led to several outbreaks in many countries worldwide, further analyses of its genetic characteristics, especially its origin and evolution, are limited, because of the lack of full-length *VP1* and complete genome sequences. Moreover, detailed studies on the genetic evolution, phylogenetic relationships, and genetic characteristics of E33 remain lacking. Therefore, we collected E33 strains from Mainland of China and sorted all E33 sequences from NCBI to fill the gaps in current research through systematic molecular evolutionary analyses.

## 2. Materials and Methods

### 2.1. Viral Isolation and Molecular Typing

Based on the surveillance of the HFMD system, seven E33 strains were isolated. Two of them were isolated from Hunan (2013HUN-019, 2013HUN-037), three from Jiangsu (2015JS-087, 2015JS-088, and 2015JS-089), and two from Xinjiang (2018XJ-113, 2018XJ-115). Their clinical manifestations include upper respiratory infection, cough, fever, runny nose, and rash on hand, foot, and mouth (Appendix A). Seven strains were cultured with human rhabdomyosarcoma (RD) cells (provided by the WHO Global Poliovirus Specialized Laboratory in the USA) and harvested when complete EV-like cytopathic effects (CPE) were observed. Viral nucleic acids were extracted using a QIAamp Viral RNA Mini Kit (Qiagen, Hilden, Germany). A one-step RT-PCR kit (TaKaRa, Shiga, Japan) was used to amplify the *VP1* region sequences with primers E33-1571F/E33-2613R (E33-1571F: TGCCTTACATCAACAGCGTC, E33-2613R: TGTGCACATGCCTAGTTTGC) and E33-2435F/E33-3273R (E33-2435F: GTACATGATGCTGTTGTTGG/E33-3273R: CCTGTTGTCATTAAATCCGC). The PCR products were purified using a QIAquick PCR purification kit (Qiagen) and then sequenced on an ABI 3130 Genetic Analyzer (Applied Biosystems, Foster City, CA, USA).

### 2.2. Whole-Genome Sequencing of E33

The whole-genome sequences of all seven strains were determined. The specific primers for E33 were designed using the NCBI online primer design website (Appendix A). The *5*′*-UTR*, *3*′*-UTR*, and *VP4* regions were amplified using primers described in other studies [23,24]. Sequencer software (version 5.4.5) was used to trim and splice the sequence fragments of E33. Similarities in the nucleotide and amino acid sequences among E33 and other *EV-B* prototype strains were compared using BioEdit software (version 7.0.9). Using the prototype strains as a reference, gene mutation analysis was performed on the coding genes of all strains in this study using BioAider software (version 1.4). 

### 2.3. Dataset Establishment

The sequences were downloaded from the NCBI database to construct datasets for genotyping, gene recombination, and genetic evolutionary analyses. First, 99 complete *VP1* sequences (843 bp) and four full-length sequences (approximately 7400 bp) were downloaded from the NCBI database as of June 2022. Together with the seven whole-genome sequences obtained in this study, 106 complete *VP1* sequences were used to classify the genotypes (Appendix A), and another poor-quality sequences were deleted. Among these reference sequences, 24 were isolated from China (22.64%) and sequences isolated from other eight countries between 1959 and 2018 were also included. Second, different regions of seven strains, including *2A, 2B, 2C, 3A, 3C,* and *3D,* were analyzed using BLAST. Twelve non-E33 sequences with similarities greater than 90% were screened as potential donor sequences. Third, through TempEst (version 1.5.3) analysis, sequences with a time bias were removed. In total, 86 *VP1* sequences were used to analyze the genetic evolution and spatiotemporal propagation paths of E33.

### 2.4. Recombination Analysis of the Seven Isolated E33 Strains

Phylogenetic trees based on the *5*′*-UTR, 3*′*-UTR, P1, P2*, and *P3* regions (including all the hydrolyzed fragments) with other *EV-B* prototypes were constructed through the neighbor-joining method in MEGA (version 7.0) using the Kimura 2-parameter model, and bootstrapping was set to 1000 bootstrap repeats. Similarity and Bootscan graphs were used to show the similarity between E33 and other *Enteroviruses B*, and a 200−nt window moving in 20−nt steps was used in Simplot software (version 3.5.1) [25]. Recombination detection program 4 (RDP4, version 4.101) was used to verify the recombinant results. RDP, GENECONV, 3Seq, Chimaera, SiScan, MaxChi, and LARD were selected for sequence analysis, at least three of which detected recombination, and a p-value less than 0.05 was considered to indicate recombination. Recombinant profiles were used to identify the potential donors of E33.

### 2.5. Phylogenetic Analysis of E33

The Markov chain Monte Carlo (MCMC) method implemented in BEAST (version 1.10, Los Angeles, CA, USA) was used for evolutionary analysis and reconstruct ancestral geographic regions [26]. Before the BEAST analysis, the uncorrected lognormal clock (UCLD) and Bayesian skyline modes were selected using the path sampling method [27,28]. The best nucleotide substitution model “SYM + G” was singled out using JModelTest (version 2.1.7) [29]. Bayesian Stochastic Search Variable Selection (BSSVS) combined with an asymmetric substitution model was used for systematic geographic analysis. Running a Markov-chain of 1 × 10^8^ steps with sampling every 1 × 10^4^ generations in BEAST. Tracer (version 1.7.1) was used to convergence checking and the calculation of the Effective Sample Size (ESS) of all estimated parameters, by discarding the first 10% of the samples. Ultimately, ensure that there were enough ESS values (greater than 200) to achieve convergence [30]. In addition, a Bayesian skyline plot was used to depict the dynamics of the population diversity in Tracer. The Bayesian maximum clade credibility (MCC) tree was constructed using Tree Annotator (version 1.10.4), with the first 10 percent of sampled trees discarded using the burn-in option [31]. The MCMC tree was visualized using Figtree (version 1.4.3), which annotates the branches according to the sampling location. The posterior probability values of different countries in the MCMC tree were extracted and plotted as a bar graph. SPREAD3 software packages (version 0.9.7) were used to calculate the Bayes factors (BF) and select important transmission paths (BF>3) to map the global spread paths of E33.

## 3. Results

### 3.1. Five Genotypes Were Reclassified Based on the VP1 Sequences

A maximum likelihood tree was constructed based on 106 E33 *VP1* sequences collected from 1959 to 2018 (Figure 1). Five genotypes (A–E) were named, with inter-group distances of 16.5–20.9% and intra-group distances of 0–14.2%. The prototype strain Toluca-3 (GenBank accession no. AJ241436.1) formed genotype A. Genotype B comprised six strains isolated from New Zealand between 1983 and 1984. Genotype C can be further subdivided into C1 and C2, with a nucleotide difference of 14.1%. The outbreak strains of New Zealand in 2000 formed sub-genotype C1, while the sub-genotype C2 included most of the Indian strains isolated from children with non-poliomyelitis acute flaccid paralysis (NP-AFP) in 2007–2008, as well as some sequences detected in Hunan, China, in 2013. In addition, five cases of NP-AFP from India and two Chinese strains were classified as genotype D during the same period. Only two Yunnan strains isolated in 2013 clustered into a single branch and belonged to genotype E. Therefore, three genotypes (B, C, and E) were circulating in Mainland of China between 2010 and 2018, among which genotype C was the dominant genotype.

### 3.2. Two Different Recombination Patterns Were Found between Different Genotypes

The full-length E33 genome was 7392 bp, with an ORF of 6551 bp that encodes a polypeptide 2183 amino acids long. There were 741 bp and 101 bp in the *5*′*-UTR* and *3*′*-UTR*, respectively. The whole-genome sequences of these seven strains had 1313 identical nucleotide changes compared to the prototype strain Toluca-3. The coding region showed 230 amino acid changes, including five non-synonymous mutations. 

Neighbor-joining phylogenetic trees were determined based on the *P1*, *P2*, and *P3* regions of all *EV-B* prototype sequences. All E33 strains clustered with the prototype Toluca-3 in the *P1* region. However, genotype C sequences were clustered with other new enteroviruses, especially EV-B88 and EV-B86 in the *P2* and *P3* regions (Figure 2), whereas genotype E was located in the same cluster with CVB6, E12, and CVB4 in these two regions. To further explore the recombination pattern of E33, we also constructed NJ trees based on the different structural and functional regions with all *EV-B* prototype sequences (Appendix A). Based on detailed phylogenetic analyses of each coding and non-coding region, the C and E genotypes clustered together with the E33 prototype strain in the *VP1*, *VP2*, and *VP3* regions. However, in other regions, these two genotypes clustered with different *EV-B* prototype strains. For instance, genotype C clustered with EV-B85, EV-B107, and EV-B74 in the *3D* region, whereas genotype E clustered with E16, indicating that the E33 recombination pattern was complex and there were many genetic variations.

Furthermore, the nucleotide similarity of genotypes C and E was compared with other *EV-B* genotypes in different sequence fragments (Appendix A). The results were consistent with those in the NJ trees. Specifically, the nucleotide sequences of the seven E33 strains were closer to Toluca-3 (76.2–79.3% similarity) than to the other *EV-B* prototype strains (54–72.1% similarity) in the *VP1*, *VP2*, and *VP3* regions. In the *P2* and *P3* regions, these seven sequences showed more homology with the other *EV-B* prototype strains than with Toluca-3, which indicated the occurrence of recombination events in a non-capsid region. Notably, the C and E genotypes were more similar to other *EV-B* prototype strains in the *VP4* region: the C genotype clustered with the E32 prototype strain, while the E genotype clustered with E3. This was also confirmed by the comparison the nucleotide and amino acid similarity in the *VP4* region between other *EV-B* prototype strains and the C or E genotype, which the nucleotide similarity was higher (79.7–84%) than E33 prototype strain (78.7–80.6%). Surprisingly, for C genotypes, the amino acid similarity between E33-2018XJ-115, E33-2018XJ-113, E33-2013HUN-19, E33-2013HUN-37, and E20 prototype strain was up to 100% in the *VP4* region, while it was 98.5% with the E33 prototype strain. As a part of the capsid protein, *VP4* is located inside the enterovirus and contains the initial codon of the enterovirus, which is closely related to viral replication and translation. The high variation in the capsid protein indicates that E33 may have the potential for rapid evolution. Therefore, this requires us to monitor them closely.

### 3.3. Special Recombination Pattern of Genotype C

Simplot and Bootscan diagrams showed that the recombination pattern of C gene subtypes in the seven strains in this study were consistent; therefore, only the results of the 2013HUN-019 strain are shown (Figure 3), others are shown in the attachment (Appendix A). The seven strains were highly similar to E13 (AB501332.1) in the 2A region; EV-B85 (JX898908.1-JX898907.1) and EV-B80 (MH614922.1, MH614924.1) in the *2B–3C* region; and CVA9 (OL519579.1) in the *3D* region. This suggests that E33 is likely to genetically recombine with the above strains in these regions. In comparison, E genotypes were isolated from Yunnan, China in 2013, whose recombination patterns were different from those of the C genotypes, which had high similarity with CVB3, E3, E4, E6, E25, and E30 in the *P2* and *P3* regions. RDP4 (version 4.101) was further used to verify the recombination donors of genotype C. The results showed that the recombination patterns of these seven strains were the same, although the breakpoints were different. Therefore, we only show four representative recombination events (Figure 4). In summary analysis, we concluded that the possible recombinant donors of the C genotype are E13, EV-B85, and EV-B80. In addition, strain CVA9, isolated in Jiangsu, China in 2018, showed high similarity with C genotype in the 3D region, which may be a potential recombinant receptor.

### 3.4. High Nucleotide Substitution Rate in the VP1 Region of E33

A maximum clade credibility (MCC) tree of 86 *VP1* sequences (843 bp) was constructed using the Bayes-Markov chain Monte Carlo method. The results showed that the average nucleotide replacement rate in the *VP1* region of E33 was 1.11 × 10^−2^ substitution/site/year (95% HPD: 8.17 × 10^−3^ to 1.4 × 10^−2^), and the most renascent common ancestor (tMRCA) originated in 1955 (95% HPD: 1945–1959). According to the Bayesian skyline analysis, we found that the genetic polymorphisms showed a stable trend until the 1990s but showed a sharp increase after the year 2000. The period from 2000 to 2012 was a relatively high plateau. Then there was another decline in the genetic diversity of the virus, which consistent with the MCC results (Figure 5). However, the results may be biased owing to the lack of sequence data.

### 3.5. Global Geographic Transmission Paths of E33

By collecting 86 complete *VP1* sequences of E33 from nine countries (Mexico, France, Thailand, New Zealand, China, India, Australia, Tunisia, and Oman), possible transmission paths of E33 were mapped globally during 1959-2018. Based on the information contained in the sequences and the root state probability distribution, we conclude that E33 probably originated in Mexico with a posteriori probability value of 0.92. Two significant transmission paths were selected by BSSVS analysis (BF >10): from Oman to Thailand (BF = 29.44) and Australia to India (BF = 15.14). In addition, E33 in Mainland of China may have been imported from Mexico (BF = 8.07) and India (BF = 5.21). Transmission paths indicate that E33 has been transmitted more frequently between Asia, Europe and Oceania (Figure 6). However, it should be noted that the BF value of the transmission path obtained in this study is not decisive supported, which suggests that more E33 genome sequences are needed to make the inferred transmission paths more convincing.

## 4. Discussion

A member of the *Enterovirus B* family, E33 is an ancient enterovirus that has caused several outbreaks. Through retrospective research, we found that the clinical symptoms caused by E33 covered a wide range of features. Although E33 can cause many severe diseases, such as aseptic meningitis and acute flaccid paralysis, it has been ignored for a long time. Domestic research on E33 is mostly limited to a single surveillance system in one province or region, so genetic evolution, recombination model and spatial transmission of E33 remain unclear. Therefore, combining E33 sequences available worldwide to predict global epidemic trends is inevitable and indispensable.

Accurate classification of genotypes is helpful to discover the differences among genotypes and further understand the genetic evolution of viruses. According to the genotype classification criteria of enteroviruses, five genotypes were determined in this study, which was inconsistent with previous reports. For instance, the virus strains previously classified as the D, E, F, and F genotypes were reclassified as the C genotype. This is probably due to the different reference sequences used and their lax partition criteria: the nucleotide differences among different genotypes were less than 15% in previous studies. Based on our reclassification, genotype C was the dominant genotype, which caused two outbreaks in previous studies. Therefore, we deduced that the viruses of the C genotype had strong transmission and virulence during their continuous evolution and variation. 

Recombination, especially intraspecific recombination, is an important evolutionary mechanism for enteroviruses [32]. As a species that contains 63 serotypes, intraspecific recombination between serotypes is more likely to occur in *Enteroviruses B* [33,34]. Simplot and RDP4 analyses showed that the genotype of E33 may recombine with EV-B80, EV-B85, CVA9, and E13 in the *P2* and *P3* regions. This was different from the recombination pattern of the Yunnan strains belonging to genotype E. Genotype C obtained in this study was more likely to recombine with the new strains of group B, whereas the Yunnan strain with genotype E was recombined with the earlier strains. Studies have shown that new enteroviruses tend to have stronger transmissibility and immunogenicity. This special recombination pattern may have led to the greater transmissibility of the C genotype, which seems to explain why the C genotype has caused two outbreaks. Unfortunately, there are only four full-length E33 sequences in GenBank, and we only carried out recombination analysis for two genotypes of C and E. More unknown recombination patterns of E33 could not be obtained.

BEAST analysis showed that E33 had an average nucleotide substitution rate of 1.11 × 10^−2^ in *VP1* region, which was faster than EV-A71 (4–7 × 10^−3^) [35], CVA16 (4.545 × 10^−3^) [36], CVA6 (4.10 × 10^−3^) [37], CVA10 (5.5–14.1×10^−3^) [38], and CVA9 (3.27 × 10^−3^) [39]. This result is consistent with the study by Lukashev et al. on the nucleotide replacement rate of non-poliovirus [40]. However, due to the lack of global surveillance of E33, the number of complete *VP1* sequences in GenBank is very limited, and the report on E33 is not sufficient, so this result may be biased by time. In spite of this, we collected as many full-length *VP1* sequences of E33 as possible, and eliminated the sequences that did not meet the time signal provided by TempEst to ensure the accuracy and reliability of the results. A high rate of nucleotide evolution suggests more rapid virus mutation and evolution, so E33 may have acquired strong environmental adaptability, high virus replication, and transmission capacity during its evolution. Trends in population dynamics showed that the population size of the virus expanded dramatically in 2000 and the nucleotide replacement rate increased accordingly. The large outbreak in New Zealand in 2000 was the main reason for the virus’ population expansion, indicating that E33 may have formed the C gene subtype and some of its inner clusters during this stage. 

Further influenced by poor surveillance and few sequences, there is currently no decisive evidence to support the geographic transmission paths of E33. Despite the insufficient sample size, we also found some international transmission paths of E33. According to the spatiotemporal propagation analysis, E33 spreads closely among Asia, Europe and Oceania. The frequent international transmission, high nucleotide replacement rate in *VP1* region, the high variation in *VP4* region and combined with the special recombination pattern of the C gene subtype, combine the above factors, E33 is likely to lead to the emergence of new genotypes in the process of evolution and there may be a risk of another outbreak. Therefore, it is necessary to strengthen the surveillance of E33 to prevent the outbreak of large-scale outbreaks. 

## 5. Conclusions

In conclusion, through this study, we have obtained a preliminary understanding of the genotypes, origin, and evolution of E33 on a global scale. However, the disease burden of E33 worldwide has been underestimated, owing to the inadequacies of the currently used surveillance systems. Therefore, this study has crucial public significance and practical utility in disease control and prevention.

## Figures and Tables

**Figure 1 pathogens-11-01379-f001:**
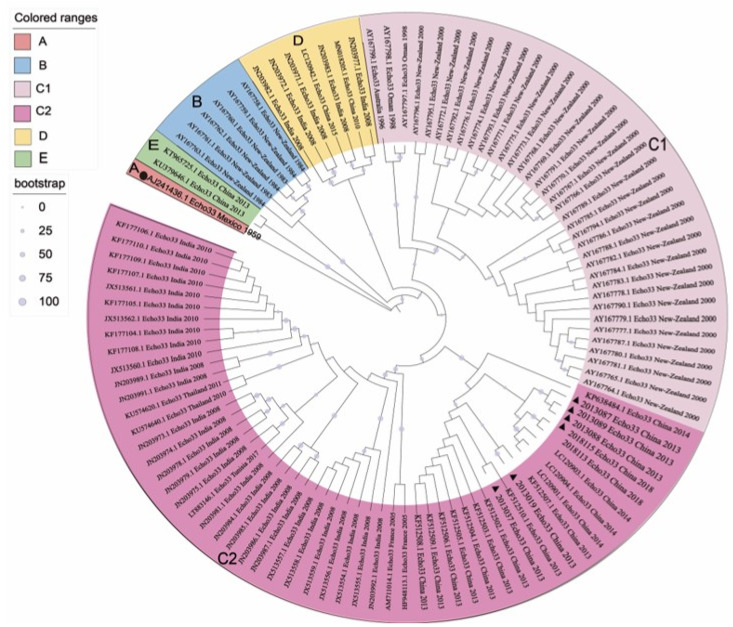
Phylogenetic tree based on complete *VP1* nucleotide sequences of E33 strains. ▲ indicates the strains in this study, ● indicates the prototype strain of E33.

**Figure 2 pathogens-11-01379-f002:**
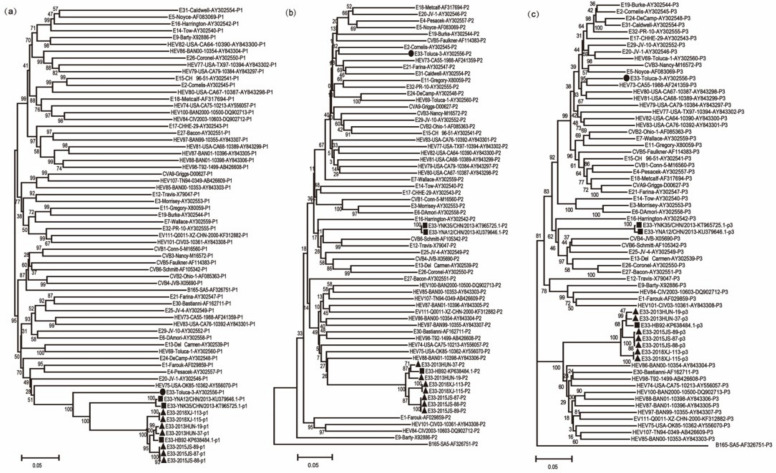
NJ trees based on *P1*, *P2*, and *P3* regions of the prototype sequence of all *Enteroviruses B* in the GenBank database and seven E33 strains in this study. ● indicates E33 prototype strain (Toluca-3); ▲indicates 7 strains of E33 in this study; ■ indicates other E33 strains in China. Numbers on codes indicate the bootstrap support of the node (1000 bootstrap replicate percentage). Scale bars represent the genetic distance, and all panels have the same scale. (**a**) *P1* coding sequences; (**b**) *P2* coding sequences; (**c**) *P3* coding sequences.

**Figure 3 pathogens-11-01379-f003:**
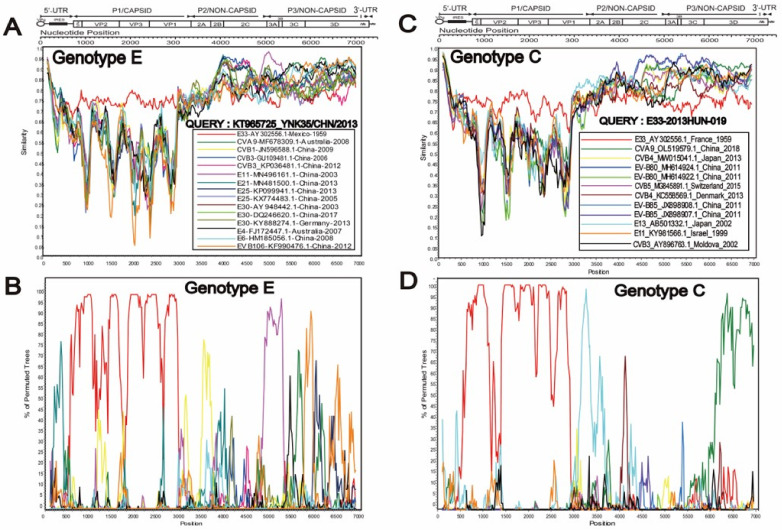
Recombination analyses of genotype E and C. **A**, **B**, **C**, **D** show separately the Simplot and BootScan plots of genotype E representing strain KT965725, as well as genotype C representing strain E33-2013HUN-019, using a sliding window of 200−nt moving in 20−nt steps.

**Figure 4 pathogens-11-01379-f004:**
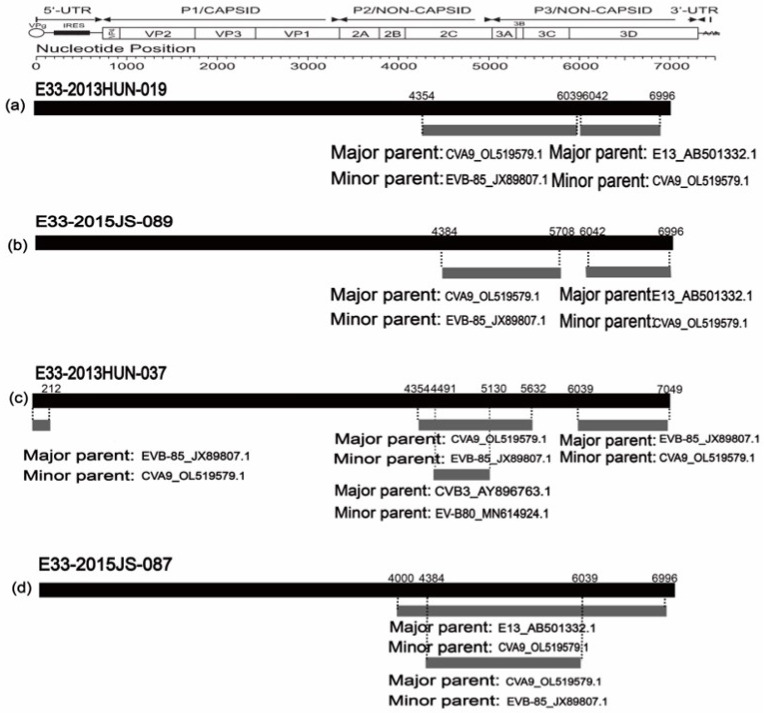
The genomic map of E33 representative strain recombination events predicted by RDP4. The black band represents the full-length genome of the E33 strains; the numbers above indicate beginning and ending breakpoint positions. The gray bands represent the recombined genomic regions; the numbers below indicate major and minor parents of the predicated recombination event. (**a**) Recombination events of E33-2013HUN-019; (**b**) Recombination events of E33-2015JS-089; (**c**) Recombination events of E33-2013HUN-037; (**d**)Recombination events of E33-2015HUN-087.

**Figure 5 pathogens-11-01379-f005:**
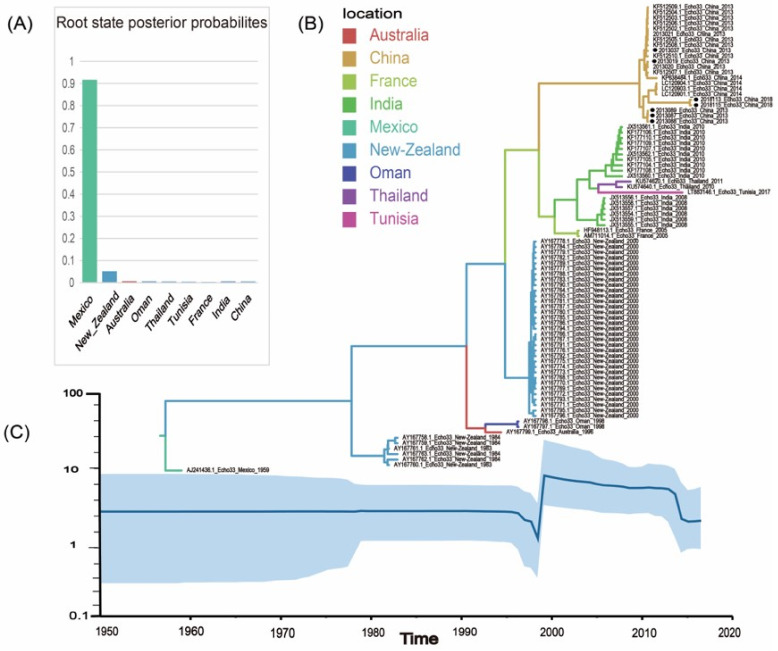
(**A**) The root state posterior probabilities estimate for each region of Bayesian phylogeographic MCC tree, which were inferred using an asymmetric substitution model. (**B**) The maximum clade credibility (MCC) phylogenetic tree generated using the Markov chain Monte Carlo (MCMC) method based on complete *VP1* sequences of 86 E33. The color of branches represents the geographical location of isolates. The solid black circles represent the sequences in this study. (**C**) Bayesian skyline plot of the E33 *VP1* region sequence, reflecting the relative genetic diversity of E33 from 1950 to 2018. The X-axis is the time scale (year), and the Y-axis is the effective population size; the solid line is the estimated median, and the blue shadow is the 95% highest posterior density.

**Figure 6 pathogens-11-01379-f006:**
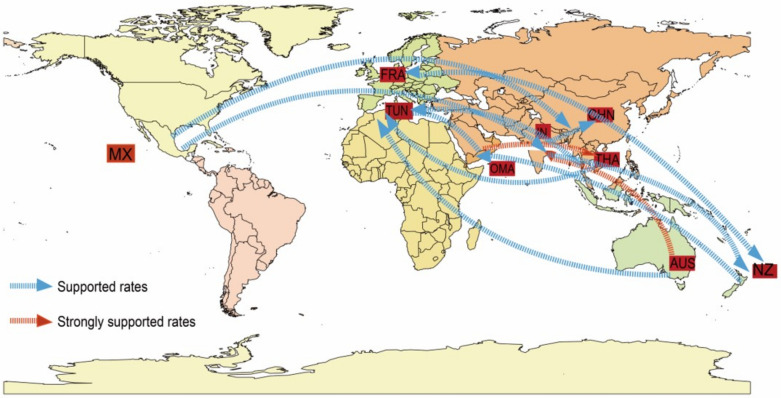
The spatial transmission route of E33. The figure only shows propagation paths that support (BF ≥ 3). The red arrow is 10 ≤ BF ≤ 100, indicating strong support rate, and the blue arrows are support rates with 3 ≤ BF ≤ 10.

## Data Availability

The complete nucleotide sequences of seven E33 strains detected in this study have been uploaded to the GenBank nucleotide sequence database; the accession number is ON873778-ON873784.

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
