# Peer review of "Molecular Characteristics and Genetic Evolution of Echovirus 33 in Mainland of China"

_pathogens, 2022, doi:10.3390/pathogens11111379_

Round 1

Reviewer 1 Report

The authors described the genotypes and evolutionary genetics of echovirus 33. E33 infection can result in serious outcomes such as encephalitis and meningitis. But it is a neglected serotype with limited reports on its origin and evolution. In this sense, results accomplished by Want et al. are valuable. The manuscript is well organized and the stated objectives are clear. Some revisions are needed.

1.      Since E33 had been associated previously with outbreaks of aseptic meningitis and upper respiratory tract infection, it will be valuable if the authors could provide detailed clinical manifestation of the seven E33 positive patients.

2.      I suggest the abbreviations of coxsackieviruses, echoviruses and newer enteroviruses refer to the literature “Recommendations for the nomenclature of enteroviruses and rhinoviruses, Archives of Virology (2020) 165:793–797”.

3.      Page 2, line 7. “Picorniviridae” should be changed to “Picornaviridae”. Besides, the taxonomy names such as “Picornaviridae” should be in italics.

4.      Page 3, line 20. The primers used to amplify the VP1 sequences should be provided.

5.      Page 3, line 42. “last”?

6.      Page 4, line 11. Mega?

7.      Page 4, line 31,32. “8” and “4” should be marked as superscripts.

8.      Figure 2 to 5. Some words are stretched in these figures, horizontally or vertically. Please maintain aspect ratio when adjusting the picture size.

9.      The authors investigated recombination events between E33 and other EV-B genomes. However, it can be observed from the similarity plot (Fig. 3) that no close genetic relationship (over >99%) can be observed in different genomic regions. So the current results demonstrated the occurrence of the recombination events but due to the limited genome data in GenBank, it is arbitrary to declare that “the main recombinant donors were E-13, CVA9, EVB85 and EVB80”. Besides, did the authors pay attention to the chronological order of so-called “donor” and the “recipient”?

10.   Page 12, line 9. Family should be changed to species.

Reviewer 2 Report

The authors have done good job in describing their study.  The interpretation of the findings and limitations of the study, however, were not comprehensive.  The submission is a presentation of genetic analysis of isolates of echovirus 33 without further evidence and link to associated factors; examples: specific location, age of the host, associated symptoms, season, among several others.  Therefore, the title is misleading through the indication “a rapidly evolving recombinant strain circulating around mainland China”.  No comparison or trend to indicate such predication.  I suggest deleting the second part of the title. The authors should be encouraged to include the limitation of this study and/or engaging infectious disease epidemiologists to analyze the demographic data associated with these isolates.  Otherwise the submission can be published as a case report of genetic characteristics of these virus strains without further extrapolation of the findings.

Round 2

Reviewer 2 Report

Thanks for considering my previous comments and suggestions.